# Fixpad++: Automated Bug Fix Verification Using LLM Agents

## Abstract

Verifying bug fixes before patches are released to end users is a critical step in the software development lifecycle. However, this process is often manual, repetitive, and error-prone, especially for crash bugs triggered through Graphical User Interface (GUI) interactions in desktop applications. Despite recent advancements in LLM-driven software agents, existing work primarily targets bug reproduction without addressing fix verification, while approaches that do focus on verification rely on source code access, making them inapplicable to closed-source GUI-based desktop applications. This paper introduces FIXPAD++, a framework designed to automatically verify bug fixes in the Notepad++ desktop application using LLM-powered agents. FIXPAD++ employs a two-phase approach: first, a multi-modal multi-agent system interacts with the buggy version to reproduce the reported crash using visual parsing and LLM reasoning. Second, upon successful reproduction, a trajectory replay mechanism executes the recorded action sequence on the patched version to validate the fix. We evaluated FIXPAD++ on FIXPAD-BENCH, a new dataset of 105 evaluation instances derived from 22 real-world Notepad++ crash bugs, including valid and invalid patches. The system achieved a reproduction success rate of 72.73% with an average time of 174.07 seconds. Among the successfully reproduced cases, FIXPAD++ correctly verified valid fixes with 87.50% accuracy and detected invalid fixes with 77.05% accuracy, outperforming OpenAI's Computer-Using Agent (CUA). FIXPAD++ demonstrates the effectiveness of specialized LLM agent architectures for automated bug fix verification in GUI-based desktop applications, offering a practical solution for automating verification workflows without requiring access to source code.

## CCS Concepts

• **Software and its engineering** → **Software testing and debugging**; *Maintaining software*.

## Keywords

Automated Bug Fix Verification, Large Language Models, Agents, Software Testing, Automated Patch Correctness Assessment, Automated Testing for Desktop Applications

**ACM Reference Format:**
Anonymous Author(s). 2026. Fixpad++: Automated Bug Fix Verification Using LLM Agents. In *Proceedings of the 3rd ACM International Conference on AI-powered Software (AIware 2026), July 06–07, 2026, Montreal, Canada.* ACM, New York, NY, USA, 10 pages. https://doi.org/XXXXXXX.XXXXXXX

## 1 Introduction

Software maintenance remains a dominant cost in the development lifecycle, estimated at 12–20% of total resources [2, 12, 28]. A critical yet often overlooked component of this phase is bug fix verification, confirming that a patch actually resolves the reported issue [30]. Without verification, bugs frequently reopen. Some studies showed that 5.3% to 13.7% of bugs are reopened, with 94% of these failures attributed to incorrect patches [3, 26, 31]. These recurring failures are costly: over 93% of reopened bugs significantly impact the regular operation of the system [20], and they take substantially longer to resolve than bugs fixed correctly the first time [20, 26].

Manually verifying a bug fix is labor-intensive. In large open-source projects like Wireshark, the average verification time is 69 days [9], and a large proportion of bug fixes remain unverified indefinitely [9, 25]. Consequently, a large proportion of bug fixes remain unverified indefinitely, reducing the overall effectiveness of bug tracking systems [9, 25]. This challenge is particularly acute for desktop applications with Graphical User Interfaces (GUIs), where traditional test automation is brittle due to the effort required to keep test scripts synchronized with evolving UI elements [8, 11, 13]; even on Android, where interaction primitives are more standardized, 44% of crashing bugs cannot be reliably replayed with existing record-and-replay tools [29].

Recent advances in LLM agents have begun to automate the reproduction side of this process. Previous research, like ReCDroid [42], LIBRO [15], and AdbGPT [10], has evolved from rule-based approaches to LLM-driven reproduction on mobile platforms, while ReBL [32] achieves high success rates through feedback-driven interaction. While frameworks like BugCraft [38] have recently brought these capabilities to the desktop domain, a significant research gap remains. Existing work primarily focuses on crash reproduction, which serves as an essential step in confirming and understanding reported bugs. However, reproduction alone does not determine whether a proposed fix actually resolves the issue. Complementing reproduction with automated verification remains unaddressed in prior work. Meanwhile, Automated Patch Correctness Assessment (APCA) techniques [33], such as FixCheck [21] and LLM4PatchCorrect [43], focus on verification but operate strictly at the code level, relying on test suites or execution traces that are often unavailable for GUI-based desktop applications and closed-source software. Practical constraints further motivate source-code-free verification as code generation models have been shown to memorize training data [1] and leak sensitive information from their training corpora [22], leading major technology companies to ban or restrict AI assistant usage for security and confidentiality reasons [16]. Currently, there is no automated solution that verifies bug fixes for desktop GUIs through visual interaction without relying on source code or internal instrumentation.

To bridge this gap, we introduce FIXPAD++, a novel framework for automated bug fix verification in GUI-based desktop applications. Unlike other approaches, FIXPAD++ operates exclusively on

visual signals. Our approach operates in two distinct phases to ensure both reliability and efficiency. First, a multi-agent reproduction system interacts with the buggy version of the application to reproduce the reported crash using visual cues. This phase comprises specialized Action, Observation, and Reflection agents that operate in a ReAct-style loop [37], augmented with self-reflection [27]. If the first phase is able to reproduce the crash, our approach records this action sequence. In the second phase, the system replays these action sequences on the patched version to assess whether the issue has been resolved. By separating the workflow into two distinct phases, we eliminate the need to re-derive the crash trajectory during the verification phase. This ensures that the fix is verified against the exact conditions that caused the failure in the reproduction phase, treating the agent's discovered path as a fixed test case. We evaluate FixPad++ on Notepad++, a representative desktop application, demonstrating that it can verify fixes without access to internal code or pre-existing test suites. Our contributions are as follows:

- **An end-to-end framework for automated GUI-level bug fix verification.** Unlike prior work that primarily focuses on crash reproduction, FixPad++ completes the verification loop by replaying crash-inducing action sequences on patched versions to determine whether reported bugs are resolved.
- **A multi-agent architecture grounded in the ReAct [37] and Reflexion [27] paradigms.** Our approach's reproduction phase has specialized agents operating in an iterative reasoning loop, enabling structured GUI interaction and self-correction.
- **FixPad-Bench, a dataset of 105 evaluation scenarios derived from Notepad++ crash bugs.** The benchmark pairs buggy instances with both correctly fixed versions and versions in which the bug persists, supporting systematic evaluation of valid and invalid fixes.

## 2 Related Work

*From Android to Desktop: The Evolution of Bug Reproduction.* Automated bug reproduction aims to convert unstructured bug reports into executable interaction sequences. Early approaches combined natural language processing (NLP) with dynamic GUI exploration; CrashDroid [35] translates crash-report call stacks into reproduction steps and replayable event traces, and ReCDroid [42] extracts steps from textual reports and replays corresponding events, achieving a 63.5% reproduction success rate on Android crashes. ReCDroid+ [41] later improved performance (e.g., 77.7%) by integrating learning-based components. The emergence of LLMs has further accelerated Android bug reproduction. AdbGPT [10] leverages prompt engineering and chain-of-thought reasoning to map natural language descriptions to GUI actions, while ReBL [32] adopts a feedback-driven interaction loop over full bug reports and execution outcomes, reporting reproduction success above 90% on Android benchmarks. A key limitation of this line of work is its predominant focus on **Android**. Android provides relatively standardized interaction primitives through its UI framework and accessibility APIs, which these reproduction systems exploit [32, 41]. In contrast, desktop environments exhibit more heterogeneous window management and interaction models, making reliable automation substantially more challenging [38].

*Visual Agents and the Verification Gap.* To cope with the complexity of desktop interfaces, agents must perceive UI elements directly from the screen. Recent vision-language approaches, such as OmniParser [18], improve structured screen parsing and UI element grounding. Crucially, these models operate solely on pixel inputs, allowing downstream agents to reason about complex visual layouts without accessing the application's source code. Building on these capabilities, multimodal agents have demonstrated general computer and smartphone control. AppAgent [39] and AutoDroid [34] focus on smartphone interaction, while ScreenAgent [23] extends planning and execution to desktop environments. Commercial systems, including Claude Computer Use [4] and OpenAI's Computer-Using Agent [24], further show that general-purpose desktop control is increasingly feasible. However, these systems target broad task automation rather than the reproducibility and correctness requirements of bug verification. BugCraft [38] targets crash bug reproduction in desktop applications, demonstrating that vision-based LLM agents can reproduce complex GUI-driven failures in Minecraft, with a reported success rate of 34.9%. To handle domain-specific interactions, BugCraft augments visual reasoning with external knowledge from the Minecraft Wiki. However, BugCraft focuses exclusively on reproduction and does not address the subsequent task of verifying whether a bug is resolved in patched versions. FixPad++ addresses this gap by reusing the *reproduction trajectory* and verifying fixes via replay-based execution on patched versions.

*Patch Correctness and the Missing GUI Interaction.* While bug reproduction focuses on triggering failures, APCA focuses on verifying fixes. Traditional APCA techniques, such as KATCH [19] and PATCH-SIM [36], rely on symbolic execution or behavioral analysis to filter incorrect patches [17]. AI-driven approaches further advance this field; FixCheck [21] combines static analysis with LLMs to generate test assertions, and LLM4PatchCorrect [43] infers correctness from execution traces. Relatedly, LIBRO [15] demonstrates that LLMs can generate bug-reproducing test cases from natural language bug reports on code benchmarks such as Defects4J. However, LIBRO operates at the code and test level rather than through GUI interaction. Moreover, recent work on LLM-based vulnerability patching has shown that code-level similarity metrics alone are insufficient for assessing patch correctness, as patches with high similarity scores still fail under execution-based validation [40]. Fundamentally, these approaches rely on access to source code and programmatic validation signals, making them inapplicable to GUI-based desktop applications that are distributed as closed-source without test suites or internal instrumentation. FixPad++ fills this gap by enabling verification based purely on observed GUI behavior, using the reproduction trajectory rather than internal code signals that raise confidentiality and intellectual property concerns [1, 22]

## 3 Methodology

### 3.1 System Overview

Figure 1 illustrates the overall architecture of FixPad++, designed to automate the assessment of patch correctness through two distinct phases: the **Reproduction Phase** and the **Verification Phase**. The pipeline takes three inputs: a bug report describing the steps

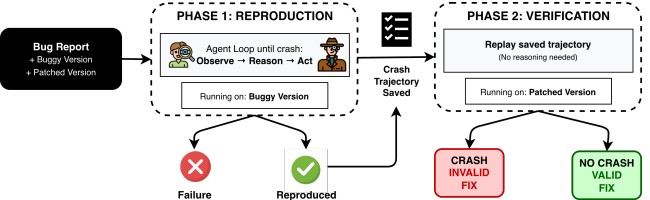

**Figure 1: Overall reproduction and verification pipelines**

to reproduce (S2R) a crash, the buggy software version where the crash occurs, and the patched software version where the fix was applied. The output is a binary verdict indicating whether the bug has been successfully fixed.

In the reproduction phase, a multi-agent reasoning system interacts with the buggy software version to reproduce the reported crash. The system iteratively reasons about the bug report and the current UI state, generates actions, executes them, and observes the results. This cycle continues until either a crash is detected or a maximum iteration limit is reached. When a crash is successfully detected, the complete sequence of executed actions, called the **trajectory**, is recorded. In the second phase, the pipeline switches to the patched software version and replays the recorded trajectory. If the same action sequence triggers a crash on the patched version, the fix is classified *invalid*; if no crash occurs, the fix is classified *valid*. By separating the reproduction phase from the verification phase, we ensure that the identified interaction sequence is genuinely crash-revealing before it is reused for verification.

## 3.2 Dataset Collection

To evaluate FIXPAD++, we required a dataset of crash bugs in a desktop application with publicly available buggy and patched versions. We initially explored several application domains with public issue trackers, including Mozilla Firefox[1], Redmine[2], and VLC media player[3]. However, these systems posed significant challenges such as multi-component instructions, unstable historical releases, and complex build pipelines, which made consistent bug reproduction difficult. Notepad++ offered a lightweight, mostly self-contained application with a stable release history, a rapid installation process, and a public GitHub issue tracker[4], making it a practical and controlled setting for our evaluation.

*3.2.1 Bug Selection.* Dataset construction began with an investigation of 8,562 closed issues from the official Notepad++ GitHub repository (as of January 15, 2026) [7]. We targeted closed issues because our verification pipeline requires the existence of both buggy and patched versions. Among the closed issues, we filtered for those containing the keyword *crash* in issue titles, body text, and labels, yielding 492 candidates. From this set we looked at each bug report and applied three inclusion criteria. Firstly, the bug must be reproducible through direct GUI interaction rather than internal or code-level defects. Secondly, the bug must not depend on third-party plugins, which introduce additional versioning and

installation dependencies, and thirdly the bug must be manually verifiable as reproducible in a buggy version and resolved in a later patched version. This yielded 22 distinct UI-driven bugs that both authors manually reproduced the buggy version and verified that they were fixed in their patched version.

*3.2.2 Dataset Augmentation.* Since our framework must not only confirm valid fixes but also detect invalid ones, we augmented the dataset with negative test cases. For each bug, we identified up to four additional Notepad++ versions in which both authors manually verified that the bug still persisted, effectively generating *invalid fix* examples. This augmentation enables evaluation of whether the system can correctly identify cases where the crash persists. While the reproduction procedure remains consistent across versions for a given bug, variations in UI layout and runtime behavior across versions introduce valuable challenges for the system. Due to limited availability of applicable older versions for two issues, the augmentation resulted in a total of 105 evaluation instances. Table 1 summarizes the dataset composition.

**Table 1: Composition of the FixPad-Bench dataset.**

| Property | Count |
|---|---|
| Crash bugs (buggy versions - distinct S2R) | 22 |
| Correctly patched versions (1 per bug) | 22 |
| Incorrectly patched versions ($\leq$4 per bug) | 83 |
| **Total evaluation instances** | **105** |

Each evaluation instance consists of a bug report, a buggy version for reproduction, and a target version (either correctly or incorrectly patched) for verification. The bug reports follow Notepad++'s structured issue template, which includes fields for Title, Description, Steps to Reproduce (S2R), Current Behavior, Expected Behavior, and Debug Information. Among these, the S2R field directly guides the reproduction agent's decision-making, while the Debug Information field provides the specific buggy version number. The corresponding patched version is manually extracted from the issue timeline, where it is typically referenced by the commit or pull request that resolved the issue.

## 3.3 Bug Reproduction Phase

Figure 2 presents the detailed architecture of FIXPAD++, including the reproduction phase. Given a bug report and the corresponding buggy software version, the reproduction phase aims to trigger the reported crash through automated GUI interaction. The system follows an extended ReAct paradigm [37] that interleaves reasoning and acting in an iterative loop. We extend this in two ways. First, we decompose the loop into specialized agents for decision-making, screen interpretation, and action evaluation. Second, inspired by the Reflexion framework [27] we introduce an explicit evaluation stage where a dedicated agent assesses whether each action achieved its intended effect and provides structured feedback to the decision-making agent, yielding a four-stage cycle: thought, action, observation, and reflection. This mirrors the iterative workflow of quality assurance (QA) engineers who act, observe, assess, and adjust.

The reproduction loop is driven by three specialized agents, **ActionAgent**, **ObservationAgent**, and **ReflectionAgent**, each

---

[1]https://bugzilla.mozilla.org/buglist.cgi?product=Firefox
[2]https://www.redmine.org/issues
[3]https://code.videolan.org/videolan/vlc/-/issues
[4]https://github.com/notepad-plus-plus/notepad-plus-plus/issues

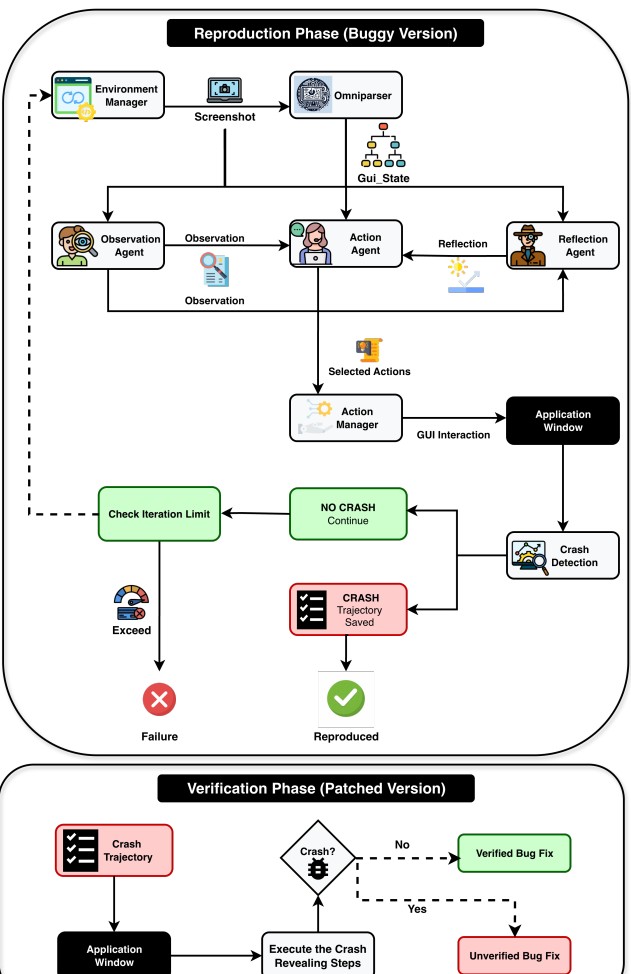

**Figure 2: Detailed reproduction and verification pipelines**

**Table 2: Prompt components provided to each agent**

| Component | Action | Observation | Reflection |
|---|---|---|---|
| Bug Report | ✓ | ✓ | ✓ |
| GUI State | ✓ | | ✓ |
| Screenshot | ✓ | ✓ | ✓ |
| Observation (NL description) | ✓ | | ✓ |
| Action Trajectory | ✓ | | ✓ |
| Reflection Feedback | ✓ | | |
| Available Actions | ✓ | | |
| Few-shot Examples | ✓ | ✓ | ✓ |

with a fixed size to ensure consistent coordinate mapping across runs. An initial screenshot is captured and parsed, and the Action-Agent makes its first decision based solely on the bug report and initial UI state. No reflection occurs in the first iteration because there is no previous action to evaluate.

*State Capture and Parsing.* The Environment Manager captures a screenshot isolated to the application window, excluding irrelevant desktop content such as the Windows menu. To enable the agents to reason about the interface, OmniParser V2[6] [18] transforms this screenshot into a **GUI State**, a structured representation of the visible UI elements. OmniParser operates solely on screenshots without requiring access to application source code, which aligns with our framework's goal of source-code-independent verification. For each captured screenshot, the GUI State provides three pieces of information via OmniParser. First, the textual content of the element is extracted through the optical character recognition capabilities of OmniParser, capturing labels such as menu names, button text, and input field contents. Second, the spatial location is recorded as a normalized bounding box that defines where the element appears on screen. Third, each element is classified as either text or icon, and flagged as interactable or non-interactable. Together, these three properties allow the agents to reason about meaningful UI elements, referencing element labels rather than pixel coordinates, while also providing precise locations for action execution. This information forms a complete GUI State. In addition to the parsed GUI State, the raw screenshot is passed as visual input to the agents, as certain interface details such as visual highlighting, color changes, or overlapping elements may not be fully captured by the parser.

*Observation Agent.* The ObservationAgent produces a natural language description of the current screen. While the GUI State provides structured element data, the agent uses natural language to capture transient visual aspects, such as which menu is open, whether text is selected, or how the layout appears. A key design choice is that the ObservationAgent receives only the screenshot and bug report, deliberately excluding the GUI State (Table 2), to force independent visual analysis that complements rather than restates the structured data. Its prompt instructs the agent to distinguish between persistent UI elements (e.g., the menu bar) and transient elements (e.g., dropdown menus), and to identify subtle but critical UI details such as selections, highlights, and fold markers. The ObservationAgent does not retain memory across iterations,

responsible for a distinct cognitive task: deciding the GUI action to be performed, perceiving the current state, and evaluating the outcome. This separation allows each agent's prompt and input to be tailored to its function. Table 2 summarizes which components each agent receives. All agents are powered by Gemini 2.5 Flash, selected for its multimodal capabilities, cost-effectiveness, and extended context window of up to one million tokens, which enables integrating bug reports, UI observations, action histories, and reflections within a single reasoning cycle. To provide deterministic output, we set the temperature to 0. To leverage in-context learning [5], each agent receives demonstrative examples that guide it in producing responses conforming to the desired structure. Complete prompt templates are provided in the replication package [5].

*Environment Manager (Initialization).* Before the loop begins, the Environment Manager installs the buggy version, launches the application, and positions the window at a fixed screen location

---

[5]https://figshare.com/articles/software/Fixpad_Replication_Package/31295002?file=61749772

[6]https://github.com/microsoft/OmniParser

**Table 3: Available actions in the ActionAgent's action space.**

| Action | Description |
|---|---|
| click(label) | Clicks a UI element specified by label. |
| move_to(bbox) | Moves the mouse to a bounding box region. |
| paste(text) | Pastes text at the current cursor location. |
| hotkey(keys) | Presses one or more keys simultaneously. |
| highlight(start_line, end_line) | Highlights lines using standard selection. |
| multi_select(start_line, end_line) | Selects lines in column mode. |
| right_click() | Right-clicks at the current cursor position. |

ensuring each description reflects only the current screen. Alongside the GUI State, this output enables the ActionAgent to plan subsequent actions and the ReflectionAgent to evaluate whether prior actions achieved their intended effects.

*Reflection Agent.* The ReflectionAgent evaluates whether the most recent action made meaningful progress toward reproducing the bug. It receives bug report, action trajectory, GUI State, observation, and screenshot and compares the ActionAgent's stated intent against the current UI to determine whether the intended effect is explicitly observable. Its prompt enforces conservative evaluation: an action is classified successful only when its effect is directly confirmed in the observation or GUI State. To decouple evaluation from decision-making, the ReflectionAgent is restricted to assessing action outcomes without suggesting next steps. This structured feedback is passed to the ActionAgent in the next iteration, enabling it to revise its strategy after a failure or proceed when progress is confirmed.

*Action Agent.* The ActionAgent, the primary decision-maker, receives the richest context among all agents (Table 2) and outputs a reasoning trace alongside one or more actions. To enable the agent to build on previous actions and avoid repeating failures, it maintains memory across iterations via the action trajectory, a running record where each entry contains a reasoning trace and the corresponding actions. This trajectory serves a dual purpose: informing future decisions during reproduction and providing the replay sequence for verification (Section 3.4).

*Action Execution.* The ActionAgent selects from a fixed action space of seven types, derived by analyzing the GUI interactions required to manually reproduce the bugs in our dataset (Table 3). This tool-calling design, as opposed to direct code execution, maintains control over the actions performed during automated interaction. To translate these semantic-level decisions into GUI events, the Action Manager employs a two-layer strategy. It first attempts to locate the target element through pywinauto[7], a Windows UI automation framework that leverages the application's accessibility tree to resolve the target label to the actual UI control. When accessibility-tree matching fails, due to custom controls not exposed in the hierarchy or ambiguous labels, the manager falls back to coordinate-based targeting using bounding box coordinates from the GUI State.

*Crash Detection and Termination.* After each action, the Environment Manager checks for crash signals. Desktop crashes appear in different forms. The system therefore monitors for three

[7]https://pywinauto.readthedocs.io/

complementary signals. An *application freeze* is detected when the application window stops responding to system messages. A *crash dialog* is identified by scanning for Windows exceptions and error windows whose titles contain crash keywords. An *unexpected termination* is detected when the application window is no longer present. Since the agent executes the bug report's reproduction steps and both authors manually observed each run, any of these signals confirms a successful reproduction. If any signal is detected, the loop terminates and the trajectory is saved. Otherwise, the loop proceeds to the next iteration. To prevent indefinite looping when handling unreproducible bugs, the interaction loop is capped at 25 iterations. In our experiments, the longest successful reproduction required 16 steps (average 6.53), making 25 a practical safety margin that accommodates complex interactions while efficiently terminating unreproducible cases.

## 3.4 Bug Fix Verification Phase

Once a crash has been successfully reproduced, the pipeline verifies whether the same behavior persists in the patched version. FIXPAD++ verifies fixes by replaying the trajectory recorded during the reproduction phase, rather than re-running the full agent loop. Each action from this recorded sequence is executed directly by the Action Manager using the same pywinauto and pyautogui mechanisms used during reproduction. This approach isolates the verification step from the initial search process, ensuring that the exact interactions that triggered the crash are repeated on the patched version. The system simply executes a known crash-inducing sequence rather than searching for it again. This makes the verification phase reproducible and faster.

The verification process, illustrated in Figure 2, proceeds through five steps. First, the Environment Manager performs a clean installation of the target patched version. We apply this process regardless of whether the fix is actually correct, utilizing the 105 instances in FixPad-Bench (Section 3.2), which contains a mix of valid and invalid fixes. Next, the application is launched and positioned exactly as it was during the reproduction phase. The recorded trajectory is then replayed action-by-action, with a fixed delay to allow for UI responsiveness. After the replay finishes, the crash detection mechanism checks for failure signals using the same three methods described in Section 3.3. Finally, the system produces a result: if a crash is detected, the fix is classified as *invalid* (bug persists); if no crash occurs, the fix is verified as *valid* (bug fixed).

## 3.5 Evaluation Setup

Our study aims to assess FIXPAD++ by answering the following research questions:

- **RQ1: Reproduction Effectiveness:** How effective is FIXPAD++ in reproducing crash-inducing bugs in buggy versions?
- **RQ2: Verification Effectiveness:** How effective is FIXPAD++ in verifying whether a bug has been correctly fixed or still persists?
- **RQ3: Baseline Comparison:** How does FIXPAD++ compare to general-purpose computer use agents in verifying crash bugs?

*3.5.1 RQ1: Reproduction Effectiveness:* To evaluate reproduction effectiveness, we run the reproduction phase (Section 3.3) on the buggy version of each crash bug in our dataset and measure whether the agent can successfully trigger the reported failure. In addition

to reporting the overall reproduction success rate, we closely inspect unsuccessful attempts to identify common failure modes and underlying causes.

*3.5.2 RQ2: Verification Effectiveness:* We evaluate verification effectiveness along two complementary dimensions: *correct fix verification* and *incorrect fix detection*. To measure Correct Fix Verification Accuracy, we replay the saved trajectory on patched versions where the bug is known to be resolved. For Incorrect Fix Detection Accuracy, we evaluate FixPad++ on versions where the bug persists despite being labeled as fixed. Since detecting incorrect fixes is crucial for preventing bug reopenings, we introduce multiple incorrectly patched versions for each bug (typically four per unique S2R) and measure how often FixPad++ correctly identifies the unresolved issue. In addition to verification accuracy, we analyze failure cases to pinpoint the primary sources of verification errors. Finally, we report verification runtime to evaluate the efficiency of our replay-based verification approach.

*3.5.3 RQ3: Baseline Comparison.* To evaluate the contribution of our multi-agent architecture reflection mechanism, we compare FixPad++ against OpenAI's CUA [24] as a baseline following the official implementation [8]. CUA represents a state-of-the-art commercial solution for GUI automation and employs a fundamentally different architectural approach: a single-agent loop that receives raw screenshots, visually interprets UI elements, and outputs actions as absolute pixel coordinates without any structured UI parsing or reflection mechanism.

We configure the baseline with a system prompt that provides the task description and S2R. To ensure a fair comparison, both systems operate on the same Notepad++ window configuration, receive the same S2R text, use the same crash detection mechanism described in Section 3.3, and run under the same iteration limit. We run CUA on both buggy versions (for reproduction) and correctly patched versions (for verification) across the same 22 bugs. We do not evaluate CUA on incorrect fix detection, as each attempt requires a full fresh agent run, making evaluation across all intermediate versions prohibitively expensive. We also re-ran the agent on cases where it failed to reproduce or verify, giving it multiple chances to succeed. Both authors manually observed all runs, documenting step-by-step behaviors and cross-version comparisons. Each bug's timestamped trajectories and detailed cost metrics are available in the replication package.

## 4 Results

## 4.1 RQ1: Reproduction Effectiveness

| Metric | Count | Rate |
|---|---|---|
| Successfully reproduced bugs | 16 / 22 | 72.73% |
| Failed reproductions | 6 / 22 | 27.27% |
| Avg. reproduction time | 174.07 seconds | |
| Avg. iterations to reproduce | 6.53 | |

**Table 4: Bug reproduction effectiveness of FixPad++.**

Table 4 summarizes the reproduction outcomes of FixPad++ across 22 crash-inducing bugs from Notepad++. FixPad++ successfully reproduced the crash behavior in 16 cases, achieving a reproduction success rate of 72.73%. These results indicate that the reproduction phase is able to follow structured S2R descriptions and interact with complex GUI workflows in the majority of cases. For the successfully reproduced bugs, the average reproduction time is 174.07 seconds with an average of 6.53 reasoning iterations per bug (ranging from 3 to 16, see replication package for details). These results indicate that the 25-iteration limit is sufficient for the reproduction phase to converge across the observe–reflect–decide–execute cycles.

Despite its overall effectiveness, FixPad++ failed to reproduce 6 out of 22 bugs (27.27%). A qualitative analysis reveals three primary root causes: limitations in low-level input simulation (2 cases), such as multi-modifier keypresses requiring precise timing; UI element recognition failures (3 cases), where small and complex elements were undetected or mismatched; and iteration limit exhaustion (1 case), where the required interaction sequence exceeded the 25-step bound. See detailed analysis in the replication package.

## 4.2 RQ2: Verification Effectiveness

| | Correct Fix Verification | Incorrect Fix Detection | Overall Verification |
|---|---|---|---|
| *All bugs* | 14/22 (63.64%) | 47/83 (56.63%) | 61/105 (58.10%) |
| *Reproduced only* | 14/16 (87.50%) | 47/61 (77.05%) | 61/77 (79.22%) |

**Table 5: Verification effectiveness of FixPad++.**

Table 5 summarizes the verification performance of FixPad++ across both correct fix verification and incorrect fix detection. Verification is performed by replaying the reproduction trajectory on the patched version, as described in Section 3.4.

*Correct Fix Verification.* Across all buggy versions, FixPad++ successfully verified 63.64% of patched versions as correctly fixed. When considering only the bugs successfully reproduced during our pipeline's reproduction phase, the verification accuracy increases to 87.50%. This indicates that, once a valid reproduction trajectory is obtained, FixPad++ is highly reliable in confirming that a bug has been fixed.

*Incorrect Fix Detection.* Detecting incorrect fixes is a critical aspect of verification, as failures in this stage may lead to bug reopenings. For each bug, we evaluate FixPad++ on multiple incorrect versions where the bug still persists (typically four per bug). Across all bugs, FixPad++ correctly rejected 56.63% of incorrect fixes. Since verification relies on replaying successful reproduction trajectories, we also report this metric conditioned on successful reproduction (prerequisite for verification), where the detection rate increases to 77.05%.

Across all verification runs, FixPad++ correctly classified 61 out of 105 versions, yielding an overall verification accuracy of 58.1%. When considering only bugs that were successfully reproduced, verification accuracy increases to 79.22% (61/77). The results show a strong association between verification performance and the system's ability to successfully reproduce the crash.

---

[8]https://github.com/openai/openai-cua-sample-app

*Verification Efficiency.* The replay-based verification design enables efficient evaluation across multiple patched versions per bug. Since the reproduction phase outputs a crash trajectory, this sequence can be reused for all subsequent verifications without repeating the costly agent process. As a result, the verification phase completes in an average of 17.59 seconds per version, compared to 174.07 seconds in the reproduction phase, resulting in faster verification. This makes the approach practical for FixPad-Bench, where each bug is evaluated against multiple candidate versions.

*Failure Analysis.* Verification failures stem from two sources. The dominant cause is reproduction failure, which accounts for 28 out of 44 failed verification attempts; bugs that cannot be successfully reproduced inherently cannot be verified. The remaining 16 failures occur among successfully reproduced bugs and are caused by UI layout differences between software versions. Because Fixpad++ replays the exact reproduction trajectory, minor layout changes such as wider menus or repositioned controls can cause actions to miss their intended targets. Each bug's verification outcomes and their failure reason are available in the replication package.

## 4.3 RQ3: Baseline Comparison

Table 6 presents the comparison between Fixpad++ and OpenAI's CUA. CUA achieved a reproduction success rate of 50.00%, compared to Fixpad++'s 72.73%. For correct fix verification, CUA reached 40.91% accuracy across all bugs, while Fixpad++ achieved 63.64%. When conditioned on successful reproduction, CUA verified 81.82% versus Fixpad++'s 87.50%. Fixpad++ outperforms the baseline across all evaluation metrics. Fixpad++ is also more cost-effective at $0.24 per bug compared to CUA's $0.29 ($0.15 for the buggy version and $0.14 for the patched version). Additionally, in two cases, CUA completed all required steps but failed to press Enter as the final action; we conservatively count these as successfully reproduced and verified. Detailed analysis, costs, comments, and trajectory logs for all CUA runs are available in the replication package.

Qualitative analysis of CUA's failure cases, compared against Fixpad++'s behavior on the same bugs, reveals three weaknesses in CUA's behavior. First, CUA could not perform column selection, which several bugs require; it attempted to approximate this through low-level mouse drags but failed consistently, whereas Fixpad++ handles this through its `multi_select` defined in ActionAgent. Second, when CUA encountered an error, such as clicking the wrong element, it had no mechanism to detect the failure and continued executing subsequent steps on an incorrect state; in contrast, Fixpad++'s ReflectionAgent identifies such failures and enables the ActionAgent to revise its approach. Detailed per-bug observations of both reproduction and verification runs are available in the replication package. These patterns reflect the benefits of a specialized action space. While CUA struggles to synthesize complex interaction primitives from scratch, Fixpad++ leverages defined custom actions to reliably execute domain-specific tasks. Furthermore, our multi-agent design decouples observation, reflection, and action into specialized roles with precise context, whereas CUA's single-agent loop is forced to perceive, reason, and act within a single model call.

| Metric | Fixpad++ | CUA Baseline |
|---|---|---|
| Reproduction rate | 16/22 (72.73%) | 11/22 (50.00%) |
| Verification rate (all) | 14/22 (63.64%) | 9/22 (40.91%) |
| Verification rate (reproduced) | 14/16 (87.50%) | 9/11 (81.82%) |
| Avg. reproduction time | 174.07 sec | 159.21 sec |
| Avg. verification time | 17.59 sec | 162.58 sec |
| Avg. total time | 191.66 sec | 321.79 sec |

**Table 6: Fixpad++ and OpenAI CUA Comparison**

## 5 Discussion

*Implications for Practitioners.* Fixpad++ has several practical implications for software developers and QA teams, particularly those maintaining large, evolving desktop applications. One common challenge in issue tracking systems is that bug reports frequently contain incomplete or missing S2R [6, 42], leading to excessive manual effort in bug triage and resolution [6]. This prolongs resolution time due to repeated clarification requests. The reproduction module of Fixpad++ can serve as an automated first-pass filter, quickly attempting to reproduce the bug. If reproduction fails, it can prompt the reporter for clarification early on, reducing developer effort, shortening the process, and ensuring that only actionable, validated bug reports proceed to the next stage of triage. On the other hand, when reproduction is successful, Fixpad++ generates an action trajectory containing natural language action descriptions paired with sequential screenshots. This can be reused by developers to verify, debug, or understand the issue more effectively—thereby reducing manual effort and speeding up resolution.

Another major source of inefficiency in software maintenance is the frequent **reopening of bug reports** due to incorrect fixes. By automating the bug fix verification process, Fixpad++ can help mitigate this problem. It can be placed as a lightweight gatekeeper before deployment: if the reproduced bug is still present in the patched version, the system can automatically notify the developer for further investigation; if the issue is resolved, it proceeds with deployment. This reduces the risk of invalid fixes being accepted, improves release reliability, and minimizes user frustration caused by recurring failures.

Maintaining **automated test suites** can be impractical in many real-world projects [14]. This is particularly true for GUI-driven desktop applications, where simulating user interactions is challenging [8, 11, 13]. Even when test suites exist, they can be difficult to keep up-to-date as the application evolves, leading to gaps in coverage or false confidence in outdated tests. Fixpad++ addresses this gap by offering an automated alternative that requires no pre-existing test infrastructure or code interventions. By leveraging natural language bug reports and real-time UI observations, it enables validation of fixes through simulated user behavior. This makes Fixpad++ especially valuable for teams without formal or consistently maintained test suites, allowing them to integrate reliable bug fix verification into their development workflow.

*Implications for Researchers.* The FixPad-Bench dataset focuses on reproducible, UI-driven crash bugs in a real-world desktop application and offers a meaningful contribution to the study of automated bug reproduction and verification. Existing datasets used in patch correctness assessment often rely on test-suite-based validation. In contrast, FixPad-Bench captures visually observable

failures, making it well-suited for evaluating agent-based or multi-modal bug reproduction and verification frameworks. As a result, it can serve as a useful benchmark for researchers exploring similar LLM-driven approaches and offers a starting point for building larger datasets targeting GUI-based software testing. In addition to the dataset, the modular architecture of FIXPAD++ demonstrates how established reasoning paradigms such as ReAct and Reflexion can be instantiated for GUI-based bug reproduction by pairing them with domain-specific tools for UI parsing, interaction execution, and crash detection. Researchers targeting other desktop applications can substitute these components while retaining the same agent loop structure, making FIXPAD++ a practical reference point for future work on LLM-based testing and verification in desktop software environments.

## 6 Threats to Validity

### 6.1 Internal Threats to Validity

*UI Understanding.* Our system adopts a hybrid approach to interpret and interact with the Notepad++ interface, combining semantic inspection through Windows UI Automation with visual parsing via Omniparser V2. While this combination helps bridge the gap between LLM reasoning and GUI interaction, it introduces several limitations. Omniparser may mislabel UI elements or merge multiple interactive components into a single detection, leading to incorrect action selection. Likewise, Windows UI Automation depends on the structure of the accessibility tree; improperly exposed or inconsistently named controls may be missed or ambiguously referenced. These visual and semantic inconsistencies can cause unintended actions or execution failures, potentially affecting the reliability of both bug reproduction and bug fix verification.

*LLM Unpredictability.* Although our system uses an LLM with a temperature setting of 0 to reduce output variability, inconsistencies may still arise due to the inherently stochastic nature of large language models. Additionally, when faced with imperfect or ambiguous UI labels, often resulting from visual parsing errors, the LLM may hallucinate components or misinterpret the environment. These unpredictable responses, even under tightly controlled settings, can lead to divergent or incomplete executions that threaten the internal validity of the bug fix verification process.

*Reusing Reproduction Steps for Verification.* To minimize variability and improve efficiency, our system uses the exact same sequence of actions for both bug reproduction and fix verification. While this approach accelerates the verification process by eliminating the need to regenerate or reinterpret steps, it assumes that the UI remains unchanged between the buggy and patched versions. In practice, even minor UI modifications, such as layout shifts or renamed controls, can invalidate previously valid steps, causing false positives during verification. This trade-off between speed and adaptability introduces a risk to internal validity, particularly in cases where the visual interface evolves between versions.

### 6.2 External Threats to Validity

*Dataset Limitation.* Our evaluation is based on a dataset of manually selected bugs in Notepad++, all of which manifest as application crashes or freezes. While this constraint allows for consistent and observable evaluation outcomes, it significantly narrows the scope of applicability. The system does not currently support functional or logic-level bugs that do not result in visible breakdowns. Consequently, both the dataset and the verification mechanism are tailored to a specific subclass of bugs, limiting the generality of the findings and reducing our ability to capture broader behavioral or UI variability across different types of failures.

*Generalizability of the Approach.* Although our approach demonstrates promising results within the Notepad++ environment, its generalizability to other applications is limited by assumptions about UI structure and reliance on Windows-specific tooling. Applications with dynamic, cross-platform, or web-based interfaces may not support the same interaction model or UI parsing strategies used in our approach. However, our results provide empirical evidence that the ReAct paradigm, which has been shown to be effective in many tasks, is also applicable to desktop GUI crash reproduction and verification. Extending this approach to other desktop environments would require domain-specific adaptations, but the underlying agent loop remains applicable.

## 7 Conclusion and Future Work

In this paper, we presented FIXPAD++, a novel framework for automated bug fix verification in GUI-based desktop applications using LLM-powered agents. FIXPAD++ employs a two-phase approach: a multi-agent reproduction system built on an extended ReAct paradigm augmented with explicit self-reflection, followed by a trajectory replay on the patched version to assess fix correctness. To support systematic evaluation, we introduced FixPad-Bench, a dataset of 105 evaluation instances derived from 22 real-world Notepad++ crash bugs, which includes both valid and invalid patches. FIXPAD++ achieved a reproduction success rate of 72.73%, and among successfully reproduced cases, it correctly verified valid fixes with 87.50% accuracy and detected invalid fixes with 77.05% accuracy. FIXPAD++ significantly outperformed OpenAI's CUA, achieving higher reproduction rates (72.73% vs. 50.00%), better correct fix verification accuracy (87.50% vs. 81.82% among reproduced bugs). These results demonstrate that automated end-to-end bug fix verification at the GUI level is achievable through specialized multi-agent architectures, bridging the gap between crash reproduction and patch correctness assessment that prior work has left unaddressed. In practice, FIXPAD++ could benefit software development teams by reducing the risk of incorrect fixes reaching end users and alleviating the manual effort associated with bug fix verification, particularly for closed-source applications where code-level verification methods are inapplicable. However, the current replay-based verification remains sensitive to UI layout differences across software versions, which can cause actions to miss their intended targets when interfaces evolve between patches. Moving forward, we plan to develop adaptive replay mechanisms that tolerate interface changes and to extend FIXPAD++ to other desktop applications across diverse domains, as well as to non-crash bug types. Ultimately, the approach presented in this paper provides a strong foundation for future research for visual automated bug fix verification, advancing the reliability and efficiency of software maintenance workflows in complex desktop environments.

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
