# OpenReview forum: "Fixpad++: Automated Bug Fix Verification Using LLM Agents"
_ACM.org/AIWare/2026/Conference — AIware 2026_

### Official Review · Reviewer_D3VR · 2026-03-08

**Rating:** 4
**Confidence:** 4

**Review:**

**Strengths**

* Addresses an important gap in automated bug verification for GUI-based desktop applications

* Clear and intuitive workflow that connects bug reproduction with fix verification

* Well-structured multi-agent architecture separating action, observation, and reasoning

* Introduction of the FixPad-Bench dataset as an initial benchmark for this problem

* Comparison with a real agent baseline helps demonstrate the value of the approach

**Weaknesses**

* Evaluation is limited to a single application (Notepad++)

* The dataset includes only 22 unique bugs

* Verification performance depends heavily on successful crash reproduction

* Replay-based verification may be fragile if the GUI layout changes between versions

**Detailed Review Comments**

The paper studies an underexplored problem: automatically verifying whether a bug fix actually resolves a crash in GUI-based desktop applications without relying on source code. It addresses an important gap in automated bug verification by proposing a workflow that links crash reproduction with fix verification through GUI interaction alone. The system identifies the sequence of actions that triggers a crash and then replays the same trajectory on the patched version to determine whether the issue persists. This design is supported by a clear multi-agent architecture that separates action execution, observation, and reasoning, thereby structuring the interaction process. The paper also introduces the FixPad-Bench dataset as an initial benchmark for this task and compares the system against a baseline agent, providing some empirical evidence of the the approach’s effectiveness.

However, the evaluation has several limitations.

1. The experiments focus solely on the Notepad++ application, raising questions about how well the system would generalise to other desktop software. Different applications may have different GUI structures, interaction patterns, or automation constraints.
2. Although the dataset contains 105 evaluation instances, they are derived from only 22 unique bugs. This limits the diversity of scenarios being tested and makes it harder to assess how the system performs across different types of failures.
3. Another limitation is that the verification stage depends entirely on successful crash reproduction. If the system fails to reproduce the bug, it cannot verify the patch. This means the approach's overall effectiveness is constrained by the reproduction stage.
4. The replay-based verification strategy may also be sensitive to changes in the user interface. If the GUI layout changes between the buggy and patched versions, the recorded interaction sequence may no longer correspond to the correct UI elements.

**Presentation and Minor Comments**

The paper is generally well written and organised. However, some parts of the architecture description could be simplified (e.g., description of the Observation Agent), and the dataset could be briefly discussed earlier in the paper to improve readability.

**Summary:**

The paper introduces Fixpad++, a framework that uses LLM agents to automatically verify bug fixes in GUI-based desktop applications. The system operates in two stages. In the first stage, a multi-agent pipeline interacts with the buggy application to reproduce crashes described in bug reports using visual parsing of the GUI and language-model reasoning. Once a crash is reproduced, the sequence of GUI actions is recorded. In the second stage, this interaction sequence is replayed on the patched version of the application to determine whether the crash still occurs. The framework employs three agents: an ActionAgent that performs GUI actions, an ObservationAgent that interprets screen states, and a ReflectionAgent that guides reasoning and decision-making. These agents operate in a ReAct-style reasoning loop with self-reflection. GUI state is extracted from screenshots using OmniParser, allowing the system to interact with the application without access to its source code. The system is evaluated on FixPad-Bench, a benchmark derived from 22 crash bugs in the Notepad++ application and expanded to 105 evaluation instances that include both correct and incorrect patches. The results show that Fixpad++ reproduces crashes in 72.73% of cases, verifies fixes correctly in 87.50% of reproduced bugs, and detects incorrect fixes with 77.05% accuracy, outperforming OpenAI’s Computer-Using Agent baseline.

---

> ### Author Response · Authors · 2026-03-19
>
> We thank the reviewer for the detailed and constructive feedback. For concerns regarding the limited evaluation scope of the FixPad-Bench dataset, generalizability of Fixpad++, and the choice of Notepad++ over other applications (Section 3.2), please refer to the following paragraphs in the **global comments**:
> * **Evaluation Scope of FixPad-Bench Dataset**
> * **Generalizability of Fixpad++**
> * **Why Notepad++ Was Selected and Why Other Applications Were Infeasible, practical barriers (Section 3.2)**.
> ### Other Comments:
> ### **Reproduction Dependency**:
> We agree that the effectiveness of the verification stage is inherently constrained by the success of the reproduction stage. In our implementation, this design choice was intentional, since verification cannot be performed reliably without first reproducing the bug: the system must observe the sequence of actions that triggers the issue. Without this step, the agent might incorrectly assume that the bug has been addressed even if the executed actions were insufficient or incorrect. This is particularly important, since detecting incorrect fixes helps prevent bug reopenings, which is one of the key motivations highlighted in the introduction. Importantly, this also mirrors standard practices in software testing: human quality assurance engineers also reproduce bugs before verifying that patches work, ensuring that the testing procedure truly exposes the issue. Our framework follows this principle to maintain the reliability and integrity of automated verification.
>
> ### **Sensitivity to UI changes:**
> We thank the reviewer for highlighting this limitation. We acknowledge that replay-based verification can be sensitive to GUI layout changes, and we discuss this in the threats to validity. In our experiments, such cases affected 15.2% of verification runs. However, this design choice was made to avoid duplicating LLM costs during verification, making the approach significantly more efficient. We plan to explore low-cost methods to improve robustness against layout changes in future work.

---

> > ### Comment · Reviewer_D3VR · 2026-03-19
> >
> > Thank you to the authors for their thoughtful responses and for addressing our concerns. I appreciate the revisions made during the rebuttal period, which have satisfactorily resolved my questions. The limitations of the work are now clearly acknowledged in the paper. Based on these improvements, I am happy to increase my score to Accept.

---

### Official Review · Reviewer_K9jp · 2026-03-11

**Rating:** 3
**Confidence:** 4

**Review:**

## Strengths

### 1. Addresses an Important but Underexplored Problem

Most prior work focuses on bug localization, patch generation, or bug reproduction, while bug fix verification remains underexplored. This paper clearly motivates the need for automated verification.


### 2. Novel Use of LLM Agents for GUI Interaction

The framework integrates several agent-based interaction. This multi-modal approach allows the system to understand bug reports and interact with GUIs in a human-like way.

### 3. Clear Presentation

This paper is well-written and easy to follow.

### 4. Practical Evaluation

The evaluation using real crash bugs in Notepad++ demonstrates real-world feasibility rather than relying solely on synthetic benchmarks.

## Weaknesses

### 1. Limited Evaluation Scope

The evaluation focuses only on Notepad++, which raises concerns about generalizability. Other desktop applications may have different GUI frameworks.
The method might struggle with complex UI behaviors or dynamic interfaces.
A broader evaluation across multiple applications would strengthen the claims.

### 2. Reproduction Dependency

The verification phase depends heavily on successful bug reproduction. If the LLM agent fails to reproduce the bug initially, the verification step cannot proceed. This introduces a potential bottleneck.

### 3. Stability of LLM-driven Interaction

LLM-based GUI interaction can be non-deterministic and sensitive to: UI layout changes, rendering differences, ambiguous bug descriptions, and LLM decoing randomness. The paper could better analyze robustness and reproducibility.

### 4. Missing Baseline Comparisons

The evaluation would be stronger with comparisons to: automated regression testing frameworks. Without baselines, it is difficult to quantify improvement.

### 5. Terminology Issue

It appears that the problem studied in this paper is more closely related to bug validation rather than bug verification. The authors may need to clarify the terminology and ensure that the terms are used consistently throughout the paper.

**Summary:**

This paper proposes Fixpad++, an automated framework for verifying bug fixes in GUI-based desktop applications using Large Language Model (LLM) agents. The authors focus on the challenge of bug fix verification, an often overlooked stage in the software maintenance lifecycle where developers confirm whether a patch truly resolves a reported issue. Traditional verification methods are typically manual, time-consuming, and error-prone, especially for crash bugs triggered through GUI interactions. Existing automated approaches either focus on bug reproduction rather than verification or rely on source code access, limiting applicability to closed-source software.

Fixpad++ addresses these limitations with a two-phase LLM-agent system: (1) Bug Reproduction Phase, (2) Multi-modal LLM agents interact with the buggy application, and (3) Visual parsing and reasoning enable the agents to navigate the GUI and reproduce crash scenarios described in bug reports.

The system is evaluated on Notepad++ crash bugs, demonstrating that Fixpad++ can successfully reproduce crashes and verify whether patches resolve them. The results suggest that LLM agents can effectively automate a previously manual verification step.

---

> ### Author Response · Authors · 2026-03-19
>
> We thank the reviewer for the detailed and constructive feedback. For concerns regarding the limited evaluation scope of the FixPad-Bench dataset, generalizability of Fixpad++, and the choice of Notepad++ over other applications (Section 3.2), please refer to the following paragraphs in the **global comments**:
> * **Evaluation Scope of FixPad-Bench Dataset**
> * **Generalizability of Fixpad++**
> * **Why Notepad++ Was Selected and Why Other Applications Were Infeasible, practical barriers (Section 3.2)**.
> ### Other Comments:
> ### **Reproduction Dependency**:
> We agree that the effectiveness of the verification stage is inherently constrained by the success of the reproduction stage. In our implementation, this design choice was intentional, since verification cannot be performed reliably without first reproducing the bug: the system must observe the sequence of actions that triggers the issue. Without this step, the agent might incorrectly assume that the bug has been addressed even if the executed actions were insufficient or incorrect. This is particularly important, since detecting incorrect fixes helps prevent bug reopenings, which is one of the key motivations highlighted in the introduction. Importantly, this also mirrors standard practices in software testing: human quality assurance engineers also reproduce bugs before verifying that patches work, ensuring that the testing procedure truly exposes the issue. Our framework follows this principle to maintain the reliability and integrity of automated verification.
>
> ### **Baseline Comparison With Automated Regression Testing Frameworks**:
> We thank the reviewer for this suggestion. We did not use automated regression-testing frameworks as baselines because they rely on predefined test cases that must be manually created and maintained, whereas Fixpad++ operates directly from bug reports without pre-existing test infrastructure. In addition, browser-oriented frameworks such as Selenium, Cypress, and Playwright are designed for web automation rather than native Windows desktop GUIs, while desktop automation tools such as Ranorex and TestComplete still depend on manually scripted tests and commercial licenses, with reported starting costs of roughly `$`4,650/year and `$`7,000/year, respectively. For these reasons, we selected OpenAI’s CUA as the baseline, since it is the closest comparable method for screenshot-based, LLM-driven desktop GUI interaction.
>
> ### **Bug Verification vs. Bug Validation Terminology:**
> We thank the reviewer for pointing out this detail. All bugs in our dataset are valid reported bugs, and our task is to reproduce these bugs and verify whether the corresponding fixes resolve them. That is why we used bug verification. This usage is also consistent with established terminology in bug tracking systems (e.g., Bugzilla's VERIFIED state) and with prior work by Souza and Chavez [30], who use "verification" for the same task. We acknowledge that in line 801, there is an overlap between the terms validation and verification. We fixed it for the camera-ready version.

---

> > ### Comment · Reviewer_K9jp · 2026-03-19
> > **Response to Authors**
> >
> > Thanks for the auhtors response, which addressed most of my concerns.

---

> > > ### Author Response · Authors · 2026-03-20
> > > **Additional Clarification on Stability of LLM-Driven Interaction**
> > >
> > > Thank you for the positive follow-up. We would like to add a brief clarification regarding the concern about the **Stability of LLM-driven Interaction.** We set the temperature to 0 to reduce output variability, but we acknowledge that this does not guarantee full determinism or reproducibility. We have discussed the related factors in the Threats to Validity (Section 6.1) and will consolidate and expand this discussion in the camera-ready version to address robustness and reproducibility more clearly.

---

### Official Review · Reviewer_JqHy · 2026-03-11

**Rating:** 3
**Confidence:** 3

**Review:**

## Significance

Strength:
- Fix verification on GUI applications where source code is unavailable is a relevant problem, and has received less attention compared to automated patch correctness assessment when source code is available.

Weakness:
- Given that the paper only focuses on Notepad++, the application of the technique is narrow. Though the paper mentions in Section 3.2 that this is due to the challenges in setting up other subjects, the technique would be more convincing if evidence on other subjects can be shown.


## Originality

Strength:
- This paper proposes a combination of reproduction and verification. Though this can be commonly done by creating reproduction tests and executing them, performing this on GUI applications by saving and replaying the trajectory seems novel.

Weakness:
- The contributions in the multi-agent design are mostly in composing the system, since the individual parts such as ReAct-style loop and self-reflection have already been extensively studied.


## Clarity

The writing is clear and easy to follow.

However, there are a few aspects which could be made clearer:
- In the methodology, there is an assumption that the verification by replay can only work if the UI elements and layout stay the same between the two versions. This was mentioned in Treats to Validity. However, it would be clearer to mention this when the technique is introduced, so that the readers can have a more holistic view of the technique before going into the evaluation results.
- Section 3.3 mentions that the Environment Manager detects crashes by checking for freeze, crash dialog, and unexpected termination. However, it’s unclear how the Environment Manager can know whether the crash corresponds to the same bug in the report. If a different bug is triggered during the reproduction, how is this detected?

Other minor issues:
- Line 77: there is repetition of the phrase “remain unverified indefinitely”.
- Line 395: “To provide deterministic output, we set the temperature to 0.” A temperature of 0 does not guarantee deterministic output from the LLM.


## Evaluation

Strength:
- Evaluation contains both correct fix verification and incorrect fix detection, which are useful in understanding the effectiveness of the technique.

Weakness:
- The evaluation is focused on only Notepad++ and only 22 bugs were considered.
- Even within the Notepad++ subject, it could be possible to evaluate on Notepad++ issues with no verification labels. This evaluation expands the number of bugs and helps to understand whether Fixpad++ can detect invalid patches that have been wrongly merged into the project.
- The reproduction phase consists of multiple agents, but there is no evidence in the evaluation on how much the multi-agent design is contributing. It would be better to have an ablation study comparing Fixpad++ with just a single agent performing the thought->action->observation->reflection altogether.

Other questions:
- Both the cost and time of CUA (line 727 and Table 6) seem to suggest that CUA has to be run twice for the reproduction and verification. Why is this the case? Can the trajectory produced from CUA’s reproduction stage be used directly for verification?

---

Overall, the paper is clear, and the combination of reproduction and verification through replay on GUI applications seems novel. However, my main concerns are on the limited subjects and bugs in the evaluation, as well as missing evidence in the contribution of multi-agent design.

**Summary:**

This paper presents Fixpad++, a framework designed to automatically verify bug fixes in the Notepad++ desktop applications. Fixpad++ uses a two-phase approach consisting of reproduction and verification. In the reproduction phase, a multi-modal multi-agent system (Observation, Reflection, Action) is used to reproduce the reported crash and generate a trajectory. In the verification phase, the previously generated trajectory is replayed to validate the fix on the patched version.
On Fixpad-Bench which consists of 22 bugs and 105 evaluation instances, Fixpad++ correctly verified valid fixes with 87.50% accuracy and detected invalid fixes with 77.05% accuracy.

---

> ### Author Response · Authors · 2026-03-19
>
> We thank the reviewer for the detailed and constructive feedback. For concerns regarding the limited evaluation scope of the FixPad-Bench dataset, generalizability of Fixpad++, and the choice of Notepad++ over other applications (Section 3.2), please refer to the following paragraphs in the **global comments**:
> * **Evaluation Scope of FixPad-Bench Dataset**
> * **Generalizability of Fixpad++**
> * **Why Notepad++ Was Selected and Why Other Applications Were Infeasible, practical barriers (Section 3.2)**.
>
> ### Questions:
> ### **CUA cost/time suggests it was run twice (line 727 and Table 6), Can its trajectory be reused?**:
> We thank the reviewer for this important question. CUA was used as a baseline for FixPad++’s reproduction and verification (RQ3). We ran CUA separately for verification because, unlike Fixpad++, its official setup does not provide a built-in trajectory-saving and replay mechanism. Fixpad++ explicitly records the successful reproduction trajectory and reuses it during verification. As a result, a successful CUA reproduction run cannot be directly reused for replay-based verification in the same way. We agree that this distinction should have been stated more clearly. We also acknowledge that combining CUA reproduction and verification costs and time does not provide a fair comparison. We will fix Line 727 and Table 6 to report them separately in the camera-ready version.
>
> ### Other Comments:
> ### **Multi-Agent Design Contribution: How much does the multi-agent design contribute over a single-agent approach?**
> We acknowledge that a direct ablation of the multi-agent design would strengthen the evaluation. While we did not include such an ablation, our RQ3 comparison with OpenAI’s CUA was intended to provide a closely related comparison point, since CUA is a state-of-the-art single-agent computer-use baseline. In that comparison, Fixpad++ achieved a 72.7% reproduction success rate, compared to 50.0% for CUA. A key difference between CUA and Fixpad++ is our multi-agent design, which separates tasks across specialized components. For example, Fixpad++’s ReflectionAgent can identify unsuccessful actions and enable the ActionAgent to revise its strategy.  As shown in RQ3, this allowed Fixpad++ to recover from some failures, whereas CUA continued execution after unsuccessful actions. This highlights the ReflectionAgent's contribution. We acknowledge that this single-agent comparison and its relation to Fixpad++ should have been emphasized more clearly in RQ3, and we will revise this in the camera-ready version.
>
> ### **Earlier Disclosure of the Replay Assumption (the suggestion to move the UI-consistency assumption into methodology)**:
> We acknowledge that the replay assumption should have been introduced earlier in the methodology, where the replay mechanism is first described, rather than introducing it in the Threats to Validity section. We moved this assumption to the methodology for the camera-ready version.
>
> ### **How does crash detection confirm the same bug, not a different one?**:
> Two of the authors manually reproduced all 22 bugs during dataset construction, which provided a concrete reference for the expected failure behavior before evaluation. During evaluation, two of the authors manually observed all runs across all the instances, and we did not observe any case in which a different bug was triggered. In addition, the Environment Manager’s crash detection is used within an S2R-guided and manually validated reproduction process, which made unrelated crash detections unlikely in our evaluation. We will clarify this procedure in the camera-ready version.
>
> ### **Minor Issues: Line 77 repetition, Line 395 temperature determinism**:
> Thanks for your detailed comments. We will fix the repetition in Line 77. For Line 395, we will replace the phrase “To provide deterministic output” with “To provide more deterministic output”, which more accurately reflects the effect of this setting.

---

> > ### Comment · Reviewer_JqHy · 2026-03-21
> >
> > Dear Authors,
> >
> > Thank you for the detailed response. Most of my concerns are addressed by the explanation and the proposed revision. I have improved my score.

---

### Author Response · Authors · 2026-03-19
**Common Concerns**

**We sincerely thank all reviewers for their constructive and detailed feedback. We provide our responses by grouping related feedback thematically to address all points in a cohesive manner.  In this section, we address the common concerns.**

### **Evaluation Scope of FixPad-Bench Dataset**:
Although our current evaluation is limited to a single application, the reproduced bugs in our benchmark largely rely on generic Windows GUI interactions rather than narrow Notepad++-specific actions. Based on our action-distribution analysis for all successfully reproduced bugs (already available in the replication package at FixPad/results/action_distribution_during_reproduction.png), over 96% of the executed actions are application-agnostic GUI actions such as click, move, hotkey, paste, and right-click, and reproducing a bug requires 14.0 actions on average. This shows that the dataset includes non-trivial GUI interaction and action sequencing common across many Windows Desktop applications, rather than simple application-specific behavior. In other words, the GUI interaction patterns of these bugs are largely common Windows desktop operations, such as opening menus, using right-click context menus, and navigating dialog boxes. We acknowledge that this evidence should have been presented more clearly, with action distribution data in the paper, and we will add this in the camera-ready version.

### **Generalizability of Fixpad++:**
Fixpad++ is designed around common interaction patterns across Windows desktop applications rather than Notepad++-specific logic. It relies on screenshot-based GUI understanding together with standard Windows GUI control tools (pywinauto and pyautogui), instead of application internals. This makes the core reasoning, perception, action, and replay pipeline centered on interface-level behavior that is shared by many GUI-based desktop applications. Accordingly, although our evaluation is limited to Notepad++, the framework is designed to be adaptable to other Windows GUI-based desktop applications. We acknowledge that this applicability could have been explained more clearly in Section 6.2, and we have clarified it in the camera-ready version.

### **Why Notepad++ Was Selected and Why Other Applications Were Infeasible, practical barriers (Section 3.2):**
Notepad++ was selected as the evaluation subject because it satisfied the practical requirements, including a publicly accessible issue tracker with reproduction information and reliable installation methods for historical releases.  This made it feasible to set up precise environments required for accurate bug reproduction and fix verification. The other applications we explored did not offer the same combination of properties. Firefox crash reports are often tied to unstable Nightly builds that evolve rapidly and are not reliably installable for historical versions, making exact environment recreation difficult. VLC crashes often depend on specific malformed media files or streaming sources that are not systematically available across versions. [Redmine (link)](https://www.redmine.org/projects/redmine/wiki/RedmineInstall) is a self-hosted web application, and most bugs depend on a particular server configuration, database state, and plugin set, making it challenging to replicate the environment for reliable evaluation. We acknowledge that this rationale could have been explained more clearly in Section 3.2, and we have revised the paper accordingly for the camera-ready version.